

# An ensemble approach based on transformation functions for natural gas price forecasting considering optimal time delays

Faramarz Saghi and Mustafa Jahangoshai Rezaee

Faculty of Industrial Engineering, Urmia University of Technology, Urmia, Iran

## ABSTRACT

Natural gas, known as the cleanest fossil fuel, plays a vital role in the economies of producing and consuming countries. Understanding and tracking the drivers of natural gas prices are of significant interest to the many economic sectors. Hence, accurately forecasting the price is very important not only for providing an effective factor for implementing energy policy but also for playing an extremely significant role in government strategic planning. The purpose of this study is to provide an approach to forecast the natural gas price. First, optimal time delays are identified by a new approach based on the Euclidean Distance between input and target vectors. Then, wavelet decomposition has been implemented to reduce noise. Moreover, fuzzy transform with different membership functions has been used for modeling uncertainty in time series. The wavelet decomposition and fuzzy transform have been integrated into the preprocessing stage. An ensemble method is used for integrating the outputs of various neural networks. The results depict that the proposed preprocessing methods used in this paper cause to improve the accuracy of natural gas price forecasting and consider uncertainty in time series.

## INTRODUCTION

In most financial matters, the current data is usually affected by past data modeled in time series, such as time series of natural gas price. These past data must be reliable to obtain acceptable results. For example, in a sample of time series, the observation value of $t-1^{st}$ must be specified to predict the value of $t$. Also, the observation value of $t-2^{st}$ must be specified to predict the value of $t-1^{st}$ and so on. The observation value of $t-2$ is called a time delay for $t-1^{st}$ period with 1 step.

The contributions of this study are divided into three parts. First, a new approach is presented to identify optimal time delays. In previous studies, the delays were identified based on the experience of experts. On the other hand, gas price is influenced by political, economic, social, etc., factors that include uncertainty in values. In the second contribution, the fuzzy transform with different membership functions is proposed to model the uncertainties in time series. Finally, in the third contribution, the wavelet

Corresponding author
Mustafa Jahangoshai Rezaee,
m.jahangoshai@uut.ac.ir

decomposition and fuzzy transform are combined for improving the accuracy of forecasting and uncertainty modeling that this type of integration has not been provided in the previous studies. Integrating the mentioned proposed algorithms with various artificial neural networks (ANNs) are used to forecast time series of natural gas price.

The rest of the paper is organized as follows: A literature review of forecasting in different areas is provided in "Literature Review". The methodologies used in this paper are presented in "Methodology". "Proposed Approach" provides the proposed approach and related subsystems. "Data and Results" presents a case study, data, results, and related analyses. "Limitations" highlights the limitations of the proposed approach in real-world forecasting problems. Finally, the summary and conclusion of the research are given in "Summary and Conclusion".

## LITERATURE REVIEW

Different kinds of financial time series have recently become a popular field for researchers. In the following, some studies in financial areas are presented. A new approach for integrating RW and ANN was proposed that employs the advantages of RW and ANN. The RW model was performed for the linear section, and the rest of the data or nonlinear section was modeled by feed-forward ANN and Elman ANN. The results show which models are better for forecasting in terms of accuracy and performance (*Adhikari & Agrawal, 2014*). In another investigation, the various machine learning methods have been used for modeling and forecasting the BUX stock index. The results showed that the generalized autoregressive conditional heteroscedasticity (GARCH) model is better than the other ones (*Marcek, 2018*). An exponent back propagation neural network has been improved to forecast the cross-correlation between two financial data (*Mo, Wang & Niu, 2016*).

*Heng-Li & Han-Chou (2015)* proposed a new method for forecasting of Nasdaq Composite Index, which is based on integrating phase space reconstruction (PSR), empirical mode decomposition (EMD), and neural networks optimized by particle swarm optimization (PSO). The empirical results have represented which the introduced PSR-EMD-NNPSO model has provided the best forecasting stock index. *Hota, Handa & Shrivas (2018)* provided a robust forecasting model using the RBF and error backpropagation network (EBPN) for BSE 30 and INR/USD Foreign Exchange. The results showed the RBF network is better than EBPN. *Wang & Wang (2017)* introduced a new approach to integrating EMD with a stochastic time-strength neural network (STNN). The mentioned method has been used to measure the forecasted results of NYSE, DAY, FTSE, and HSI stock market indexes. *Kusy (2018)* proposed a new algorithm for the structure reduction of the probabilistic neural network (PNN). The calculations have been applied fuzzy C-means (FCMs) for choosing PNN's pattern neurons based on the derived centroids. *Hidayat et al. (2016)* developed a neural network model and applied it for inflation data in Indonesia. They optimized the development of a feed-forward neural network model. In the following, some studies are presented about the integration of wavelet decomposition with ANNs.

*Bai et al. (2016)* integrated wavelet decomposition with a backpropagation neural network (BPNN) for improving the accuracy of air pollutants forecasting. Wavelet decomposition was applied to decompose original data to different levels. Then, each level was considered input of BPNN. *Yu et al. (2018)* proposed a hybrid approach for short-term wind speed forecasting by the wavelet packet decomposition (WPD), density-based spatial clustering of applications with noise (DBSCAN), and the Elman neural network (ENN). The results showed the WPD-DBSCAN-ENN model was better than the others. *Ji et al. (2014)* introduced an approach based on the wavelet neural network for short-term forecasting of the parking guidance information systems problem. Also, an integration wavelet decomposition and ANN are presented for daily streamflow forecasting. The results showed WA hybrid models are better than classic ANNs (*Guimarães Santos & Silva, 2014*). In another research, *Dadkhah, Rezaee & Chavoshi (2018)* provided an approach integrating clustering and classification algorithms for predicting short-term power output forecasting of hourly operation based on climate factors. Also, they examined the effects of wind direction and wind speed on the accuracy of prediction results. In the following, some studies about using fuzzy models for forecasting time series are presented.

An adaptive fuzzy inference system was proposed for predicting the power plant output by considering various climate factors (*Rezaee, Dadkhah & Falahinia, 2019*). The authors have been applied metaheuristic algorithms for increasing the accuracy and performance of the designed system. In another study, a hybrid model has been used for chaotic time series prediction. An approach has been proposed for training second-order Takagi- Sugeno- Kang fuzzy systems using an adaptive neuro-fuzzy inference system (ANFIS) (*Heydari, Vali & Gharaveisi, 2016*). A similar study for forecasting time series using a robust fuzzy time series model may be found in *Yolcu & Lam (2017)*. In another research, for forecasting air pollution in Turkey, a novel fuzzy time series model has been used based on the Fuzzy K-Medoid clustering (*Dincer & Akkuş, 2018*). The proposed method handled outliers in time series. *Wang, Li & Lu (2018)* applied fuzzy logic to forecast air pollution in China. They combined the fuzzy time series forecasting method and data reprocessing approaches for forecasting main air pollutants.

Using new artificial neural networks are another category of a forecasting method. *Wang et al. (2018)* introduced a semi-heterogeneous approach to forecast crude oil prices. The mentioned approach applied decomposed techniques, including wavelet decomposition, singular spectral analysis, empirical mode decomposition, and variation mode decomposition in the preprocessing stage. The autoregressive, ARIMA, support vector regression (SVR), and ANN have been used to forecast. Finally, the results of forecasting have been integrated by a new combination approach. *Agrawal, Muchahary & Tripathi (2019)* provided a new ensemble approach based on relevance vector machines (RVM) and boosted trees. In the first stage, the RVM has been used based on Gaussian RBF and polynomial kernels. The performance of the model has been improved by Extreme Gradient Boosting. The outputs of each model have been integrated using Elastic net regression. *Rezaee, Jozmaleki & Valipour (2018)* provided an approach to predict the stock market using dynamic fuzzy C-means, Data Envelopment Analysis, and

multilayer perceptron (MLP). In another study, a novel ensemble method has been introduced for time series forecasting integrating various machine learning models (*Adhikari, 2015*). Finally, the MLP has been integrated with inverse neural network and mathematical programming for determining optimal daily amounts of fuel consumption used by a power plant (*Rezaee & Dadkhah, 2019*).

## METHODOLOGY

### Wavelet transform

In computational and functional analysis, a discrete wavelet transform (DWT) is a general WT for which the wavelets are discretely sampled. In this paper, a type of DWT family known as Daubechies with order 18 (Daubechies-18) has been used in the preprocessing stage (*Han, Kamber & Pei, 2012*).

$$a = 2^j \ \ j \in Z \ \text{ and } \ b = ka \ \ \ \ k \in Z \tag{1}$$

where $a = 2^j$ is the scale Parameter and $k$ is the time-translation parameter. Therefore, the discrete wavelet is expressed as follows:

$$\phi_{j,k}(t) = \ 2^{\frac{-j}{2}} \ \phi\left(2^{-j}t - k\right) \tag{2}$$

It gives the range of $j$ and $k$ as $0 < k < 2^{J-j} - 1$ and $1 < j < J$.

### Fuzzy transform

The basic idea of fuzzy transformation is to transform an original space into a special space for discovering information. It can use a continuous function on a fixed distance, $[a, b] \in R$ (transforming the fuzzy space of functions on an open space of real numbers by a direct transformation into the real n-dimensional vectors). Their inverse conversion returns the derived n-vector to the initial or approximate function of it.

Let $x_1 < x_2 < \ \cdots < x_n$ be fixed nodes in $[a, b]$, such that $x_1 = a, \ x_n = b$ and fuzzy sets $A_1, \ A_2, \ \ldots, \ A_n$ have their membership functions $A_1(x), \ A_2(x), \ \ldots, \ A_n(x)$ defined on $[a, b]$, form a fuzzy partition if they satisfy the following conditions ($k = 1, \ 2, \ \ldots, \ n$) :

1. $A_k \ : [a, \ b] \ \rightarrow [0, 1], \ A_k(x_k) = 1,$

2. $A_k(x_k) = 0 \ if \ x \notin ( \ x_{k-1} \ , \ x_{k+1})$ where $x_1 = a$ and $x_n = b$

3. $A_k(x)$ is continuous,

4. $A_k(x), \ k = 2, 3, \ \ldots n$ are strictly increasing on $[x_{k-1}, \ x_k]$ and
   $A_k(x), \ k = 1, 2, \ \ldots, \ n - 1$

5. $\forall x \ \ x \in [a, \ b] \ \sum_{k=1}^{n} A_k(x) = 1 \tag{3}$

6. $A_k(x_k - x) = A_k(x_k + x), \ \text{for all } x \in [0, \ h], \ k = 2, \ \ldots, \ n - 1$ where all
   $x_1, \ x_2, \ \ldots, \ x_n$ are equidistant, i.e.,
   $x_k = a + h(k - 1), \ k = 1, \ \ldots, \ n, \ h = (b - a)/(n - 1).$

7. $A_k(x) = A_{k-1}(x - h)$, For all $k = 2, 3, \ldots, n - 1$ and $\in [x_k, x_{k+1}]$ , and $A_{k+1}(x) = A_k(x - h)$, for all $k = 2, \ldots, n - 1$ and $x \in [x_k, x_{k+1}]$ where $h = (b - a)/(n - 1)$.

where $h$ is the length of support of $A_1 \ldots A_n$ and $2h$ is the support of other basic functions $A_k$, $k = 2, 3, \ldots, n - 1$ in case of uniform partition. The membership functions $A_1, A_2, \ldots, A_n$ are called basic functions.

Let $C([a, b])$ be the set of continuous functions on the domain $[a, b]$, $A_1, A_2, \ldots, A_n$ are basic functions on $[a, b]$ and let $f$ be any function in $C([a, b])$. Then $n$-tuple vector of real numbers of $[F_1, F_2, \ldots, F_n]$ is defined by Eq. (4):

$$F_k = \frac{\int_a^b f(x) A_k(x) dx}{\int_a^b A_k(x) dx} \; , \quad k = 1, 2, \ldots, n. \tag{4}$$

Equation (4) is called the F-transform $f$ concerning $A_1, A_2, \ldots, A_n$. In case of discrete F-transform, $f$ be given at nodes $P_1, P_2, \ldots, P_l$ and $A_1, A_2, \ldots, A_n$ , $n < l$ are basic functions which form a fuzzy partition on $[a, b]$, then, $[F_1, F_2, \ldots, F_n]$ are given by the following formula:

$$F_k = \frac{\left( \sum_{j=1}^l f(P_j) A_k(P_j) \right)}{\left( \sum_{j=1}^l A_k(P_j) \right)} \tag{5}$$

One of the most important issues in fuzzy systems is membership functions. These functions specify the membership value of variables. They can be defined as either discrete or continuous. If a defined variable to be is a discrete variable, the discrete membership function is used. If the variable being defined is continuous, the continuous membership function is defined. For continuous variables, membership functions may be either mathematical or real analytic functions. The fuzzy transform functions used in this research are provided in Table 1.

## Multilayer perceptron

MLP is the most famous ANN applied in time series forecasting problems. A multilayer perceptron is essentially a feed-forward structure of the entrance, one or more hidden layers, and one output layer. There are some nodes on each layer that are linked through irreversible links to a forwarding layer. The relationship between inputs and outputs in a multilayer perceptron network is given by the following formula:

$$y_t = G\left(a_0 + \sum_{j=1}^h F\left(\alpha_j \beta_{0j} + \sum_{i=1}^p \beta_{ij} \times y_{t-i}\right)\right) \tag{10}$$

where $\alpha_j$ and $\beta_{ij}$ $(i = 1, 2, \ldots, p; j = 1, 2, \ldots, h)$ are the alliance weights, $\alpha_0$ and $\beta_{0j}$ are the bias, and F, G are the hidden and output layer activation functions, respectively. In the MLP, each input data $(y_1, y_2, y_{t-p})$ is first multiplied in relative weights $(\beta_{11}, \beta_{21}, \beta_{p1})$, then it is added by $\beta_{01}$, multiple in $\alpha_j$ and enters into the transfer function (F). The output of the transformation function is found by multiplying the weights with their corresponding inputs, adding the bias $(\alpha_0)$, and entering into the activation function (G).

**Table 1 The membership functions used for considering uncertainty in time series.**

| | | |
|---|---|---|
| Triangular membership function | $\mu(x) = \max\left(\min\left(\dfrac{x-a}{b-a}, \dfrac{c-x}{c-b}\right), 0\right)$ | **(6)** |
| Trapezoidal membership function | $\mu(x) = \max\left(\min\left(\dfrac{x-a}{b-a}, 1, \dfrac{d-x}{d-c}\right), 0\right)$ | (7) |
| Gaussian membership function | $\mu(x) = \exp\left[-\dfrac{1}{2}\left(\dfrac{x-m}{\sigma}\right)^2\right]$ | (8) |
| Bell-shaped membership function | $\mu(x) = \dfrac{1}{1 + \left\lvert\dfrac{x-m}{\sigma}\right\rvert^{2a}}$ | (9) |

This calculation will keep on until $j = h$. $h$ is the number of hidden layer neurons. The most common activation function in the MLP is the sigmoid function.

## Radial basis function

RBF operates radial-base function as the activation function. The output of this network is a linear mixture of radial-base functions for input parameters and neurons. RBF networks are used in function approximation, time series forecasting, classification, and control of the system. RBF networks have a three-layer structure. The initial layer is described as the input layer, the next layer is described as the hidden layer, and the latest layer is described as the output layer. The RBF describes the potential of $jth$ neuron in the layer on the ground of varying of the Euclidean distance obtained by the vector:

$$u^j = \lVert x - w^j \rVert^2, \text{ for } j = 1, 2, \ldots, s \tag{11}$$

where $x$ is the $k$-dimensional input vector, and $w^j$ represents the weights in the hidden layer. The RBF neural network uses varying species of activation functions famous on the ground of Gaussian or RBF. The activation function for $jth$ hidden neuron is described as:

$$\psi_2(u^j) = e^{\frac{-u^j}{2\sigma_j^2}} = e^{\frac{\lVert x - w^{j^2}\rVert}{2\sigma_j^2}}, \text{ for } j = 1, 2, \ldots, s \tag{12}$$

where $\sigma_j^2$ is the variance of the inward time series. $\lVert x - w^j \rVert$ is the Euclidean norm, $w^j$ and $\sigma$ stand for the center and variance of Gaussian function, respectively. $x = \{x_1^p, x_2^p, \ldots, x_m^p\}$ is the $p$th input vector. If the time series data in the entrance vectors are not erect, hence the activation function makes Eq. (13).

$$\psi_2(u^j) = e^{\frac{-(x-w^j)^T \sum^{-1}(x-w^j)}{2}} \tag{13}$$

where $\sum^{-1}$ is the reverse of the variance-covariance matrix for the entrance vectors of time series.

## Group method of data handling

Group method of data handling (GMDH) has some of the layers, and each layer has a number of neurons. All neurons utilize contiguous structures in which have two entrances

and one output. Without loss of generality, suppose for each neuron, five weights, and one bias to bring about processing function between input and output time series. Therefore, we have:

$$y_{ik}^* = N(x_{i\alpha},\ x_{i\beta}) = b^k + w_1^k x_{i\alpha} + w_2^k x_{i\beta} + w_3^k x_{i\alpha}^2 + w_4^k x_{i\beta}^2 + w_5^k x_{i\alpha} x_{i\beta} \tag{14}$$

where $(i = 1,\ 2,\ 3,\ \ldots,\ N)$, $N$ is the number of input and output data $(K = 1,\ 2,\ 3,\ \ldots,\ C_m^2)$, $\alpha,\ \beta \in \{1,\ 2,\ 3,\ \ldots,\ m\}$, $m$ is the number of the preceding layer weights computed by MSE and therefore obvious and stable values are substituted in each neuron. The number of candidates of new neuron foundation is calculated by $C_m^2 = \frac{m(m-1)}{2}$.

The GMDH algorithm is capable of constructing models for complex systems with a high degree of regression. The criterion for selecting the optimal neural network model is to minimize the relationship (15). A model with less prediction error is considered a desirable model. In the relationship (15), $n$ is the number of observations.

$$E = \frac{\sum_{i=1}^n \left(y_i^{forecast} - y_i^{actual}\right)^2}{n} \tag{15}$$

## Adaptive neuro-fuzzy inference system

ANFIS was developed in 1990. Fuzzy Inference System (FIS) is based on the nonlinear approach that delineates the input/output relationships of an actual system using a set of if-then rules. In this neural network, the Takagi-Sugeno species FIS has been used. The output of each rule is a linear mixture of entrance variables. The ultimate output is the average weighted output of each rule. Theoretically, ANFIS models include five primary sections: Input and output database, a fuzzy model creator, an FIS, and a consistent ANN. ANFIS eliminates the initial problem in the design of the fuzzy system, the parameters of the membership function, and the design of the if-then rules. The ability to train ANNs are used for creating automated fuzzy rules and parameters optimization.

## Support vector regression

Support vector machines (SVMs) were first used to solve classification problems. Then, it was developed for regression problems in 1992. Support vector machines associated with modeling and prediction are called support vector regression. In the SVR, the activation functions are first mapped on the input vectors, and the kernel functions perform (.) product between the outputs of these mapped vectors. The original SVR formulation is:

$$f(x) = w\phi(x) + b \tag{16}$$

where $f(x)$ is the evaluated model output, $w$ is a weight vector, and $b$ is a bias period. The vector $w$ is a component of the characteristic space of the issue. Hence the issue is to select the straight nonlinear maps $\phi(x)$, and the fitting for excellence RBF is one of the

grossest used nonlinear kernels for SVR models and is performed in this paper. In SVR, first, the input support vectors $(x_1, x_2, x_N)$ and $x_{test}$ are considered inputs of the activation functions. Then, activation functions are mapped on the input vectors. The output of the activation function for each input vector is multiplied by the output value of the $x_{test}$ activation function, and each result is multiplied by the corresponding weight $(\alpha_1, \alpha_2, \ldots, \alpha_N)$ and enters into the transformation function. The output of the transformation function in the form of a linear combination is aggregated with a bias $(b)$, and the output is derived from the SVR model.

## Ensemble forecasting model

Preceding studies have shown which combination models commonly lead to distinct forecasts. An ensemble method is referred to as multiple forecasts and handles linear or nonlinear integrations of the forecasting results, direct to a coherence forecast. This research checks an ensemble method to combination forecasting of natural gas price, where the forecasts are established from five models, including decomposition and transformation techniques. If there are $m$ types of forecasting models for solving a special problem, the outcomes of them are added together. Suppose that:

$$\text{Combination}_{model}(\text{Model} = \text{MLP}, \text{RBF}, \text{GMDH}, \text{ANFIS}, \text{SVR})$$

where $\text{Combination}_{model}$ is the forecast combination of each model, the output of the forecast combination model based on five individual models can be displayed as:

$$P = f(y_1^{forecast}, y_2^{forecast}, \ldots, y_m^{forecast}) \tag{17}$$

*Chu et al. (2006)*, according to a study at Stanford University, computed and evaluated the complexity of various Machine Learning algorithms on Multicore CPUs. Table 2 shows the complexity of the algorithms used in this paper based on the mentioned study (updated in some cases). The complexities may be changed according to the number of layers, the number of neurons, epochs/iterations, or other hyperparameters of each algorithm.

Where $m$, $n$, $p$, and $l$ are the number of data/batches, the number of neurons in each layer, number of features, and the number of rules, respectively.

## PROPOSED APPROACH

Time-series data of natural gas price are gathered, which have 3,720 records, with daily data from 2004/09/09 to 2019/06/10, which were obtained from http://finance.yahoo.com. The descriptive statistics including the measures for natural gas daily prices: Minimum = 1.49, Mean = 4.6451, Maximum = 15.39, Median = 3.86, SD = 2.3406, Skewness = 1.5064, Kurtosis = 5.3989. Time delays play an important role in increasing forecasting accuracy. Determining the optimal time delay has a great impact on the training process of models in time series forecasting. If optimal time delays are not selected, the training process is hampered, and neural networks perform the learning process with low accuracy. On the other hand, overfitting occurs when the network does the learning process more than necessary and the network is not able to forecast appropriately new data.

**Table 2 Complexity analysis of algorithms used in this paper.**

| Machine learning algorithm | Complexity |
|---|---|
| MLP (three layers with backpropagation algorithm for each iteration) | $O(mpn^3)$ |
| RBF | $O(mpn^2)$ |
| GMDH (for maximum three layers) | $O(mpn^2+pn^3) \sim O(n^3)$ |
| ANFIS (using fuzzy c-means algorithm and for each iteration) | $O(mpl)$ |
| SVR | $O(m^2p)$ |

For handling this problem, the models should adjust hyper-parameters. The values of all hyper-parameters such as learning rate, epochs, the number of inputs membership functions in ANFIS, the number of clusters in ANFIS, max layer neurons and alpha parameters in GMDH, hidden layer size in MLP, spread and max neurons in RBF, and epsilon and sigma in SVR are determined by various methods proposed in articles about this issue. Various values are examined to obtain the best conclusions in terms of the accuracy and performance of ANNs for test data (unseen data).

Therefore, in the preprocessing stage, a new method has been proposed to identify optimal time delays for time series of natural gas price. The steps of the new approach to identify optimal time delays of time series are expressed as follows:

Step 1: Determining input and target vectors for each delay.
Step 2: Normalizing input and output vectors.
Step 3: Calculating the Euclidean distance between input and output vectors for each delay.
Step 4: Calculating the ratio between the consecutive distances in pairs.
Step 5: Determining larger ratios as optimal delays.

Then, wavelet decomposition is used to study natural gas prices and reduce noise and increase forecasting accuracy. The mentioned time series data were decomposed by wavelet decomposition into four levels as the best number of levels (*w1, w2, w3, w4*).

To model the uncertainty, the fuzzy transforms with different membership functions are used. Time-series are inherently uncertain which the fuzzy approach is used for importing uncertainty to the system. In a fuzzy system, membership functions play an important role. Therefore, in this study, five different and well-known membership functions are used in a fuzzy transform called Triangular type 1, Gaussian, Trapezoidal, Triangular type 2, and Bell-shaped. F-transform is applied for decomposing the original time series, and therefore, the decomposed data are used as inputs (*F1, F2, F3, F4, F5, F6*). Figure 1 shows the results of the fuzzy transformation with the Triangular type 1 membership function for the time series of natural gas prices. Figures 1A–1F show the different levels of inputs based on the Triangular type 1 function.

Afterward, the decomposition techniques and combinations are integrated with MLP, RBF, GMDH, ANFIS, and SVR. The efficiency and accuracy of the neural networks are evaluated by the RMSE and *R* (correlation coefficient). In summary, this study uses wavelet decomposition and fuzzy transform in the preprocessing phase. Also, MLP, RBF,

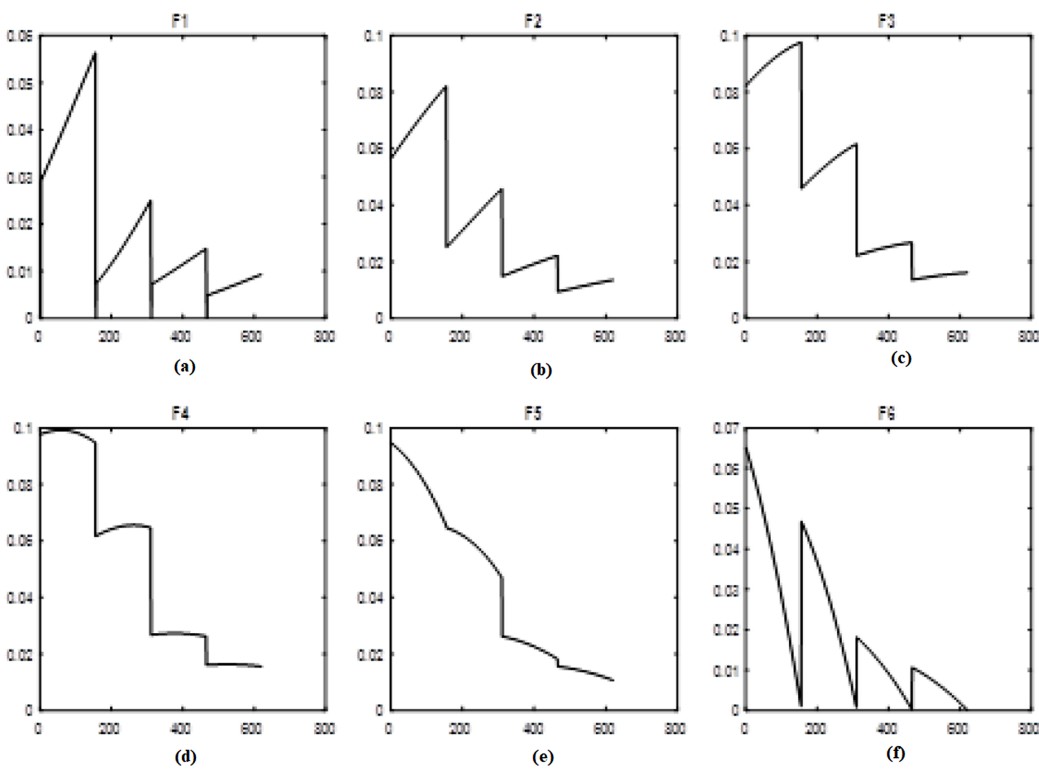

**Figure 1 Levels of fuzzy transform with Triangular type 1 membership function.** (A–F) The different levels of inputs based on the Triangular type 1 function.

GMDH, ANFIS, and SVR are applied for forecasting the natural gas price. Finally, an ensemble method combines the results. These steps are shown in Fig. 2.

## DATA AND RESULTS

Energy is supplied by various carriers such as oil, gas, etc. According to the effective role in the development and economics of countries and due to the limited resources, price forecasting is an essential tool for the short-term energy market. In this study, for all models, 70% and 30% of data are considered the train and test data, respectively. Also, optimal time delays are identified according to the new method presented in the previous section. According to the calculations, delay 1 has the largest ratio and is considered the optimal delay. Then, other delays with higher ratios are added to the delay 1 respectively, until there is no improvement in the neural network performance.

Finally, delays 1, 2, and 5 are considered optimal delays for the natural gas price time series. The identified delays are rational because the current day price is related to one, two, and five working days ago. The same delays are used in all calculations for the gas price time series.

First, the original time series is considered the input of neural networks without wavelet decomposition and fuzzy transformation, with time delays of 1, 2, and 5. In this study, the correlation coefficient ($R$) is a criterion that indicates the severity of the relationship

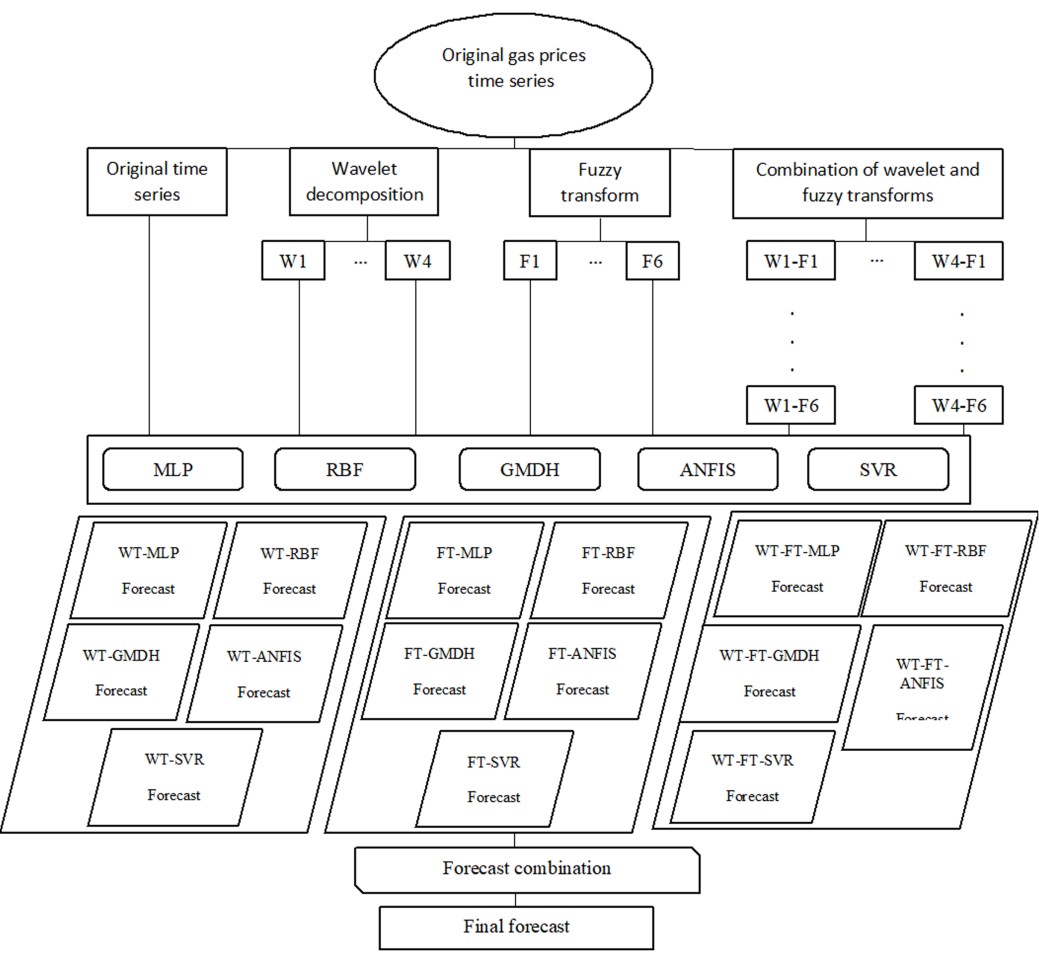

**Figure 2 Flowchart of preprocessing and forecasting steps.**

as well as the type of relationship (direct or reverse) between the two set variables (targets and outputs). According to the results, the RBF performs the best result for the original natural gas price time series. According to the results in Fig. 3, the root-mean-square error (RMSE) of the RBF for time series is 0.1483, and the correlation coefficient is 0.95731.

In the next examination, the wavelet decomposition and various neural networks are integrated.

In Table 3, the original time series of natural gas price is decomposed by wavelet decomposition into four levels. The levels (*w1-w2-w3-w4*) are considered the input of neural networks. Table 3 shows the results of implementing neural networks for four levels. The sum of noises that each wavelet level has derived from the original data is calculated. The sum of noises for Levels 1 to 4 is equal to 203.6893, 280.7320, 330.8307, and 435.7385, respectively. Also, the sum of the cumulative noise taken up to Level 4 is 1251. In Table 3, 20 combinations are evaluated. The W4-MLP combination has the best

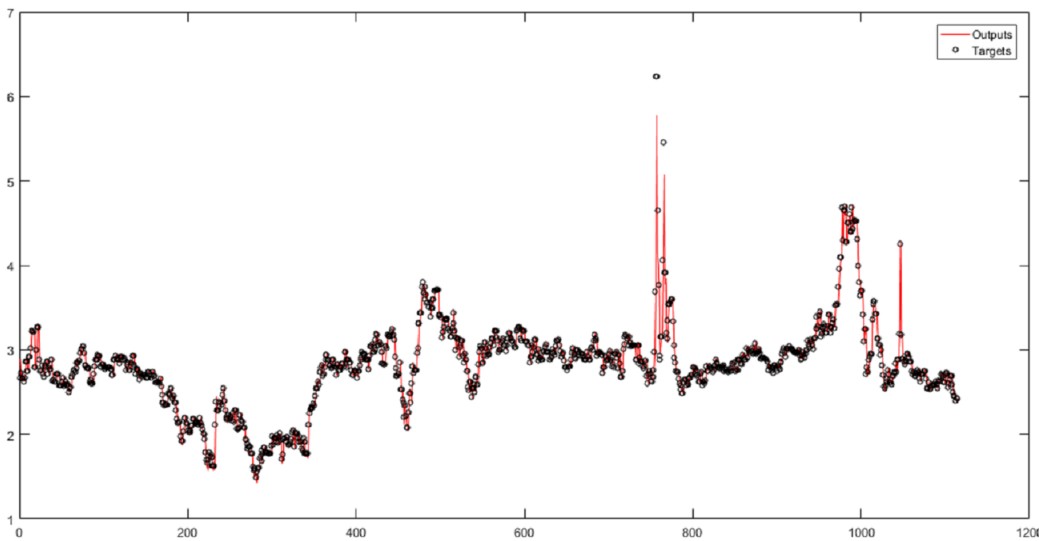

**Figure 3 The results of the RBF neural network for the original time series of gas price.**

**Table 3 Calculating error and correlation coefficient for time series applied wavelet decomposition.**

|  | RMSE (train) | R (train) | RMSE (test) | R (test) |
|---|---|---|---|---|
| W1-MLP | 0.15357 | 0.99792 | 0.082804 | 0.98659 |
| W1-RBF | 0.15355 | 0.99791 | 0.095043 | 0.98334 |
| W1-GMDH | 0.16353 | 0.99763 | 0.10148 | 0.98105 |
| W1-ANFIS | 0.15766 | 0.9978 | 0.091917 | 0.98392 |
| W1-SVR | 0.16212 | 0.99767 | 0.095822 | 0.98214 |
| W2-MLP | 0.036885 | 0.99988 | 0.021522 | 0.99906 |
| W2-RBF | 0.037009 | 0.99988 | 0.023842 | 0.99888 |
| W2-GMDH | 0.055633 | 0.99973 | 0.038534 | 0.99719 |
| W2-ANFIS | 0.038451 | 0.99987 | 0.024723 | 0.99879 |
| W2-SVR | 0.056466 | 0.99972 | 0.033676 | 0.99773 |
| W3-MLP | 0.0062521 | 1 | 0.0036305 | 0.99997 |
| W3-RBF | 0.0063061 | 1 | 0.0036619 | 0.99997 |
| W3-GMDH | 0.017222 | 0.99997 | 0.010397 | 0.99981 |
| W3-ANFIS | 0.0065844 | 1 | 0.003712 | 0.99997 |
| W3-SVR | 0.03922 | 0.99986 | 0.022998 | 0.99893 |
| W4-MLP | **0.0014284** | **1** | **0.0006167** | **1** |
| W4-RBF | 0.00125 | 1 | 0.0006467 | 1 |
| W4-GMDH | 0.0073589 | 1 | 0.0040815 | 0.99997 |
| W4-ANFIS | 0.0012957 | 1 | 0.0006323 | 1 |
| W4-SVR | 0.022275 | 0.99996 | 0.014339 | 0.99965 |

Note:
The bold cell in each column shows the best value reported for each criterion.

performance. However, its performance is very little different from the performance of other networks. According to Fig. 4, the root-mean-square error of the W4-MLP is 0.061674, and the correlation coefficient is 1.

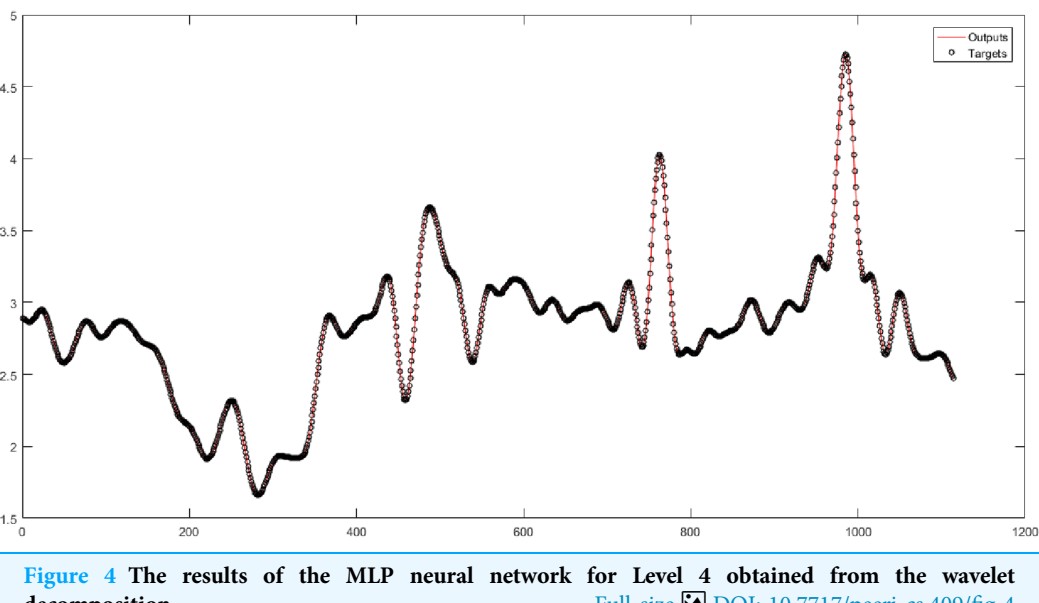

**Figure 4 The results of the MLP neural network for Level 4 obtained from the wavelet decomposition.**

For reducing noise and increasing the accuracy, the natural gas price time series was decomposed by wavelet transform into four levels. According to Table 4, it should be determined which of these levels are more similar to the original data. In this study, to examine the behavior and performance of the presented algorithms, the noise between original and decomposed data has been calculated and presented in Table 4. Calculating noise between original data and decomposed data has been used by two criteria (RMSE and R). Less root-mean-square error and high correlation coefficient mean less difference between the original data and decomposed data. Of course, it does not mean which the decomposed data inside itself has less noise and swing. Table 4 compares the calculated values by RMSE and R. It depicts W1-ANFIS has the most similarity to the original time series. In other words, the W1-ANFIS has the least noise toward the original data and has a closer behavior to it. Also, W1-MLP, W1-GMDH, and W1-RBF have similar behavior to the original time series. In the next stage, the fuzzy neural networks with different membership functions are used (see Fig. 2). According to this figure, first, the original time series is decomposed into six levels by fuzzy transform with five membership functions, then the fuzzy levels (*F1-F2-F3-F4-F5-F6*) are considered inputs of each model.

In Table 5 (second and third columns), the Triangular type 1 membership function is used. They show the results of implementing neural networks for six fuzzy levels with Triangular type 1 membership function. In these columns, 30 combinations are considered. According to the mentioned columns, the MLP has the best performance for Level 6 of the fuzzy transform. The root-mean-square error of the MLP for Level 6 is 0.00010555, and the correlation coefficient is 0.99949 (see Fig. 5).

In the fourth and fifth columns of Table 5, the Gaussian membership function is used in the fuzzy transform structure. They represent the results of implementing neural networks for six fuzzy levels with the Gaussian membership function. In these columns,

**Table 4 Calculation of noise between original data and decomposed data by wavelet transform.**

|          | RMSE     | R       |
|----------|----------|---------|
| W1-MLP   | 0.098071 | 0.98102 |
| W2-MLP   | 0.14015  | 0.9609  |
| W3-MLP   | 0.14577  | 0.95729 |
| W4-MLP   | 0.17024  | 0.94122 |
| W1-RBF   | 0.11242  | 0.97656 |
| W2-RBF   | 0.13914  | 0.96151 |
| W3-RBF   | 0.14471  | 0.95778 |
| W4-RBF   | 0.16683  | 0.94341 |
| W1-GMDH  | 0.10229  | 0.98104 |
| W2-GMDH  | 0.13709  | 0.96286 |
| W3-GMDH  | 0.14509  | 0.95724 |
| W4-GMDH  | 0.16661  | 0.94297 |
| W1-ANFIS | 0.095041 | 0.98267 |
| W2-ANFIS | 0.15258  | 0.9542  |
| W3-ANFIS | 0.15735  | 0.95065 |
| W4-ANFIS | 0.17997  | 0.93473 |
| W1-SVR   | 0.16793  | 0.94787 |
| W2-SVR   | 0.17651  | 0.94237 |
| W3-SVR   | 0.17981  | 0.93844 |
| W4-SVR   | 0.18461  | 0.93287 |

30 combinations are evaluated. According to the results, the SVR has the best performance for Level 1. According to Fig. 6, the root-mean-square error of the SVR for Level 1 is 0.0002834, and the correlation coefficient is 0.95448.

In the sixth and seventh columns of Table 5, the Trapezoidal membership function is used in the fuzzy transform structure. They depict the performance of neural networks for six fuzzy levels with the Trapezoidal membership function. In these columns, the time series is forecasted based on 30 different combinations. According to the results, the ANFIS has the best performance for Level 6. Figure 7 presents the root-mean-square error and the correlation coefficient for Level 6 with values 0.00063896, and 0.9625 respectively.

According to the eighth and ninth columns of Table 5, the Triangular type 2 membership function is used in the fuzzy transform structure. They show the results of implementing neural networks for six fuzzy levels with Triangular type 2 membership function. In these columns, 30 different combinations are evaluated. SVR has the best performance for Level 3. According to Fig. 8, the root-mean-square error of the SVR for Level 3 is 0.00016989, and the correlation coefficient is 0.95957.

According to Table 5 (10th and 11th columns), the Bell-shaped membership function is used. These columns present the results of implementing neural networks for six fuzzy levels with the Bell-shaped membership function. According to this table, 30 combinations are considered. The SVR has the best performance for Level 3 of the fuzzy transform.

**Table 5 Calculating error and correlation coefficient of time series by fuzzy neural networks with various membership functions.**

|  | Triangular Type 1 | | Gaussian | | Trapezoidal | | Triangular Type 2 | | Bell-shaped | |
|---|---|---|---|---|---|---|---|---|---|---|
|  | RMSE (test) | R (test) | RMSE (test) | R (test) | RMSE (test) | R (test) | RMSE (test) | R (test) | RMSE (test) | R (test) |
| F1-MLP | 0.00077095 | 0.96429 | 0.00029688 | 0.95079 | 0.00018713 | 0.94312 | 0.00023004 | 0.95002 | 0.000074295 | 0.97667 |
| F1-RBF | 0.00089835 | 0.95538 | 0.00047533 | 0.88392 | 0.00049873 | 0.95647 | 0.00040971 | 0.91047 | 0.00010867 | 0.93474 |
| F1-GMDH | 0.00089677 | 0.95575 | 0.00030744 | 0.88272 | 0.00034627 | 0.94506 | 0.00017237 | 0.95894 | 0.000049928 | 0.97914 |
| F1-ANFIS | 0.0008857 | 0.95544 | 0.00043059 | 0.87954 | 0.00097418 | 0.82092 | 0.0003805 | 0.87243 | 0.000072496 | 0.96111 |
| F1-SVR | 0.011021 | 0.95239 | 0.00025853 | 0.80092 | 0.00025624 | 0.94462 | 0.0002436 | 0.86729 | 0.000047595 | 0.96606 |
| F2-MLP | 0.00096726 | 0.96964 | 0.00042193 | 0.94287 | 0.00063972 | 0.96233 | 0.00027185 | 0.93324 | 0.00004843 | 0.9689 |
| F2-RBF | 0.0009696 | 0.96997 | 0.00029145 | 0.95781 | 0.00021151 | 0.94216 | 0.00025795 | 0.94839 | 0.000075494 | 0.97789 |
| F2-GMDH | 0.0011351 | 0.95551 | 0.00045547 | 0.88869 | 0.00061929 | 0.94694 | 0.00041263 | 0.91145 | 0.0001351 | 0.94137 |
| F2-ANFIS | 0.00096457 | 0.97 | 0.00036663 | 0.88684 | 0.00034359 | 0.9455 | 0.00021831 | 0.95862 | 0.000096299 | 0.98399 |
| F2-SVR | 0.020301 | 0.94807 | 0.00047876 | 0.85566 | 0.0010501 | 0.79268 | 0.00042402 | 0.85079 | 0.00007986 | 0.95711 |
| F3-MLP | 0.00094221 | 0.975 | 0.00028629 | 0.79847 | 0.00027515 | 0.93855 | 0.0002896 | 0.85124 | 0.000045736 | 0.97052 |
| F3-RBF | 0.00097379 | 0.97421 | 0.00047731 | 0.92484 | 0.00066549 | 0.96232 | 0.00027782 | 0.93014 | 0.000052228 | 0.9718 |
| F3-GMDH | 0.0026492 | 0.84415 | 0.00029649 | 0.95791 | 0.00021458 | 0.94488 | 0.00026718 | 0.95029 | 0.000079557 | 0.97704 |
| F3-ANFIS | 0.00098146 | 0.97421 | 0.00046039 | 0.88481 | 0.0005375 | 0.95597 | 0.00041016 | 0.9092 | 0.00011344 | 0.94166 |
| F3-SVR | 0.019744 | 0.9553 | 0.00036274 | 0.88229 | 0.00034244 | 0.94562 | 0.00020767 | 0.95657 | 0.000095443 | 0.98397 |
| F4-MLP | 0.00076628 | 0.98065 | 0.00047923 | 0.85738 | 0.0011875 | 0.77548 | 0.00042635 | 0.8444 | 0.000082222 | 0.95733 |
| F4-RBF | 0.00076543 | 0.98105 | 0.00032689 | 0.79973 | 0.00027352 | 0.94065 | 0.00028689 | 0.85996 | 0.000045848 | 0.97056 |
| F4-GMDH | 0.0008288 | 0.97694 | 0.00047408 | 0.93003 | 0.00067282 | 0.96138 | 0.00027152 | 0.93391 | 0.000062642 | 0.9699 |
| F4-ANFIS | 0.00076557 | 0.98104 | 0.00029412 | 0.9586 | 0.00020249 | 0.94216 | 0.00026196 | 0.9497 | 0.000075291 | 0.97789 |
| F4-SVR | 0.019265 | 0.96732 | 0.00046006 | 0.88919 | 0.00053991 | 0.95016 | 0.00041158 | 0.91247 | 0.0001027 | 0.94256 |
| F5-MLP | 0.00031118 | 0.99495 | 0.00034228 | 0.88742 | 0.00034541 | 0.94583 | 0.00021571 | 0.95863 | 0.000048891 | 0.98421 |
| F5-RBF | 0.00020388 | 0.99703 | 0.00047886 | 0.85565 | 0.0010366 | 0.79575 | 0.00042866 | 0.84917 | 0.000076752 | 0.95751 |
| F5-GMDH | 0.0013618 | 0.86725 | 0.00026898 | 0.79867 | 0.000272 | 0.93883 | 0.00026593 | 0.85346 | 0.000044912 | 0.97053 |
| F5-ANFIS | 0.00020426 | 0.99703 | 0.00047765 | 0.92553 | **0.00063896** | **0.9625** | 0.00027832 | 0.93011 | 0.000045137 | 0.97178 |
| F5-SVR | 0.02087 | 0.99477 | **0.0002834** | **0.95448** | 0.00019017 | 0.94023 | 0.00023964 | 0.94398 | 0.000081016 | 0.97305 |
| F6-MLP | **0.00010555** | **0.99949** | 0.00046531 | 0.88291 | 0.00051476 | 0.95395 | 0.0004165 | 0.90795 | 0.00010673 | 0.93262 |
| F6-RBF | 0.00097823 | 0.96968 | 0.0003128 | 0.88239 | 0.00034145 | 0.94489 | **0.00016989** | **0.95957** | **0.000046212** | **0.98144** |
| F6-GMDH | 0.0012245 | 0.96956 | 0.00050671 | 0.83456 | 0.0010225 | 0.80092 | 0.00042943 | 0.84146 | 0.000089798 | 0.94124 |
| F6-ANFIS | 0.00078845 | 0.96959 | 0.0002738 | 0.77788 | 0.00027034 | 0.93807 | 0.00026978 | 0.84247 | 0.000061622 | 0.96019 |
| F6-SVR | 0.019798 | 0.94619 | 0.00046632 | 0.92919 | 0.00063962 | 0.96284 | 0.00027712 | 0.93069 | 0.000049766 | 0.96511 |

**Note:**
The bold cell in each column shows the best value reported for each criterion.

Figure 9 shows the root-mean-square error and the correlation coefficient for Level 3 with values of 0.000046212 and 0.98144.

In the final stage, the combination of wavelet decomposition and fuzzy transformation is done. First, the original time series is decomposed by wavelet transform into four levels. Then, the levels decomposed by wavelet are considered the input of fuzzy transform with different membership functions and are subdivided into six levels. Continuing, these compounds are considered the input of neural networks.

In each wavelet-fuzzy combination with a distinct membership function, 120 different combinations are evaluated. Table 6 presents the best combination in terms of
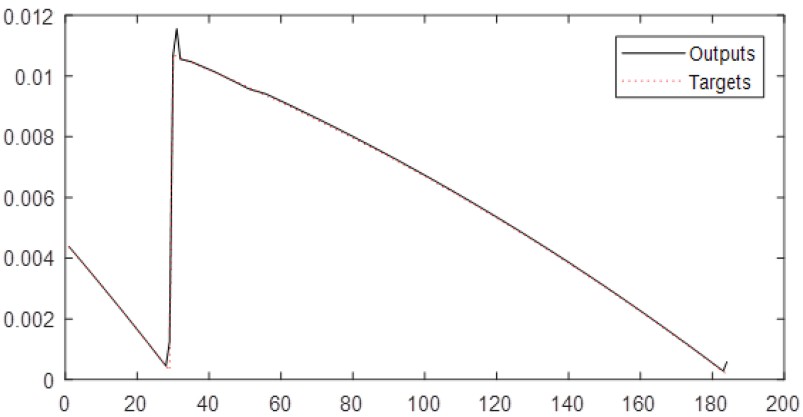

**Figure 5 The results of MLP for Level 6 obtained from fuzzy transform with Triangular type 1 membership function.**

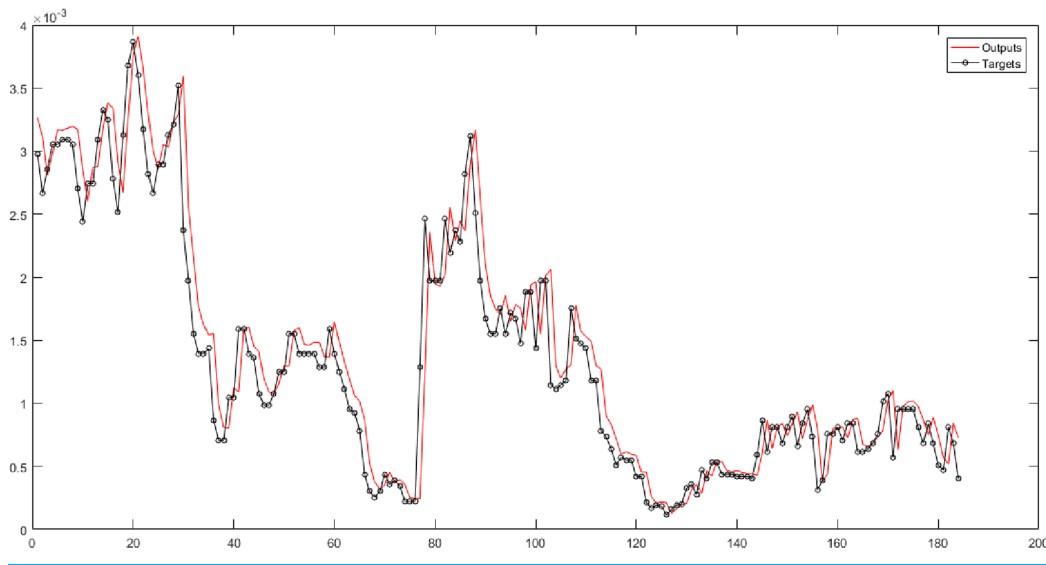

**Figure 6 The results of SVR for Level 1 obtained from fuzzy transform with Gaussian membership function.**

performance and accuracy. In this stage, 600 different combinations are made to forecast the natural gas price. In the combination of wavelet and fuzzy transform with the Triangular type 1 membership function, W1-F5-RBF is the best performing among 120 compounds. Combined with the Gaussian, Trapezoidal, Triangular type 2, and Bell-shaped membership functions, W4-F5-MLP, W2-F1-MLP, W4-F5-ANFIS, and W4-F6-ANFIS have the best performance, respectively. By comparing the results of decomposition techniques, the wavelet decomposition performed better result than others. Then, the fuzzy transform with the Triangular type 1 membership function has the best performance. Similarly, the combination of wavelet decomposition and fuzzy transform separately, the Triangular type 1 and the Bell-shaped membership functions have appropriate performance.

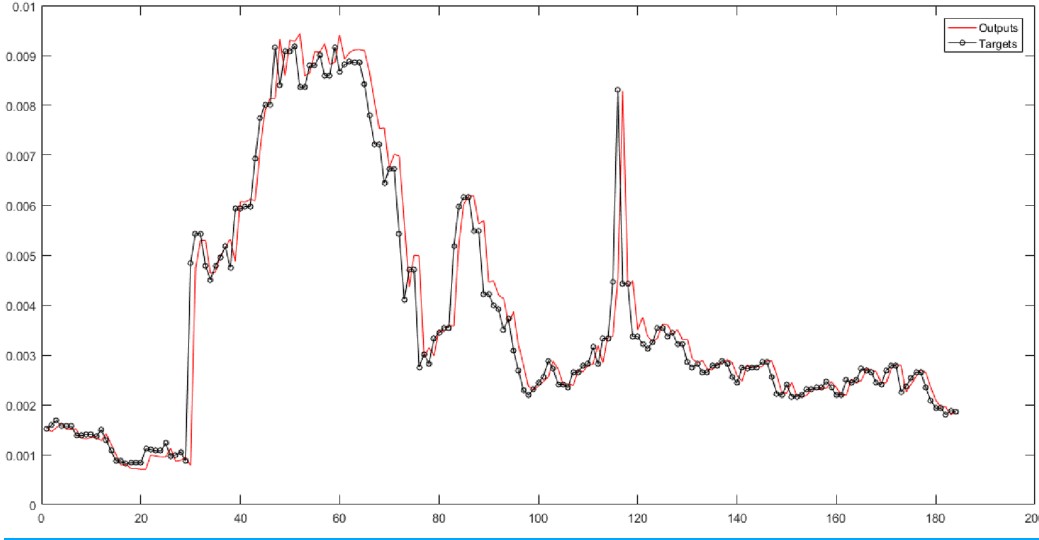

**Figure 7 The results of ANFIS for Level 6 obtained from fuzzy transform with the Trapezoidal membership function.**

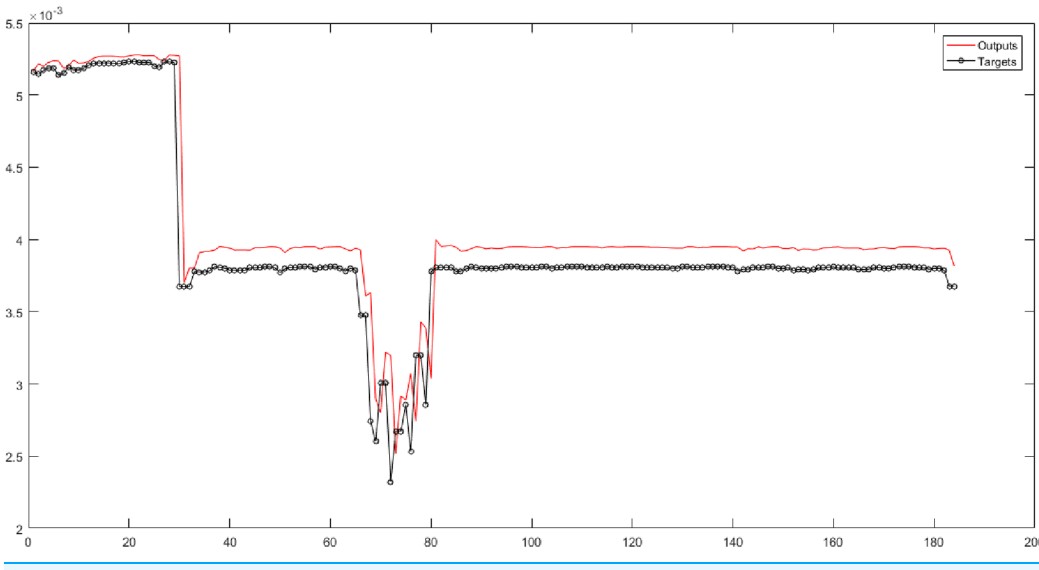

**Figure 8 The results of SVR for Level 3 obtained from fuzzy transform with the Triangular type 2 membership function.**

In Table 7, in the single model, the original time series is considered the input of neural networks. In this state, no decomposition technique is used. The results of the single model for each network are presented in Table 7. Then, decomposition techniques are applied. In DWT, the first original data is decomposed into four levels. These four levels are integrated and considered the input of neural networks. The results of the DWT are presented in Table 7. In the fuzzy transform, the first original data is considered the input of fuzzy transform with five membership functions. The original data is decomposed into six levels by the fuzzy transforms with various membership functions. Thirty levels are obtained from fuzzy transforms with five membership functions, and they are integrated

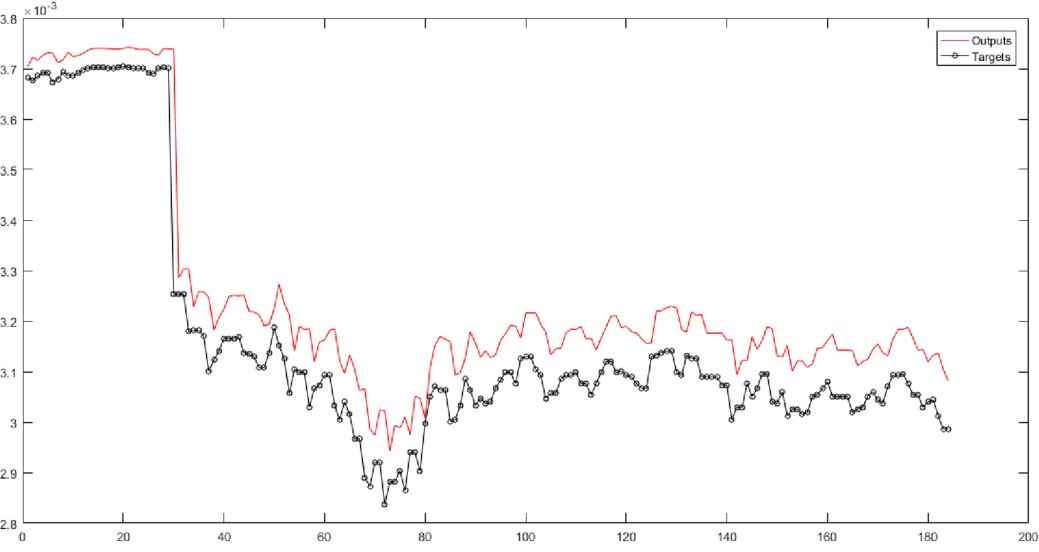

**Figure 9 The results of SVR for Level 3 obtained from fuzzy transform with the Bell-shaped membership function.**

**Table 6 Calculating error and correlation coefficient of time series by wavelet-fuzzy transformation with five membership functions.**

|  | The best performance among 120 forecasted combinations | RMSE (train) | R (train) | RMSE (test) | R (test) |
|---|---|---|---|---|---|
| WT-FT (Triangular type 1 mf) | W1-F5-RBF | 0.0011086 | 0.99898 | 0.00011825 | 0.99875 |
| WT-FT (Gaussian mf) | W4-F5-MLP | 0.00015879 | 0.99717 | 0.000020236 | 0.99811 |
| WT-FT (Trapezoidal mf) | W2-F1-MLP | 0.00041631 | 0.99816 | 0.000072099 | 0.9918 |
| WT-FT (Triangular type 2 mf) | W4-F5-ANFIS | 0.00016253 | 0.99769 | 0.000030826 | 0.99746 |
| WT-FT (Bell-shaped mf) | W4-F6-ANFIS | 0.00007106 | 0.99901 | 0.000010122 | 0.9987 |

and considered the input of neural networks. The results of the FT for each network are presented in this table. In DWT-FT, first, the original data is decomposed into four levels by wavelet decomposition. These decomposed levels are considered the input of fuzzy transforms with five membership functions. The results of DWT-FT for each network are presented in Table 7.

According to Table 8, in the single model, original data is considered input of neural networks. Then, the output of neural networks is calculated, and these output vectors are integrated. According to the obtained output and target, the root-mean-square error and the correlation coefficient are calculated. In the combined model, different levels are integrated and considered input of neural networks. Then, the outputs of neural networks are calculated and integrated. According to the obtained target and output, the root-mean-square error and the correlation coefficient are calculated (see Table 8). The results and accuracy of forecasting are improved when the decomposition techniques are used. In the single model, five levels are integrated, but in the combination model, 154

**Table 7 Results of models under combined methods.**

| | | RMSE (train) | R (train) | RMSE (test) | R (test) |
|---|---|---|---|---|---|
| MLP | Single model | 0.25641 | 0.99418 | 0.15086 | 0.95575 |
| | DWT | 0.20008 | 0.99978 | 0.11801 | 0.99818 |
| | FT | 0.0092812 | 0.99712 | 0.0027558 | 0.98336 |
| | DWT-FT | 0.084845 | 0.98665 | 0.0090304 | 0.983 |
| RBF | Single model | 0.2601 | 0.99401 | 0.1483 | 0.95731 |
| | DWT | 0.19436 | 0.99979 | 0.12927 | 0.99779 |
| | FT | 0.0093718 | 0.99706 | 0.0029492 | 0.98242 |
| | DWT-FT | 0.039197 | 0.9971 | 0.010384 | 0.97874 |
| GMDH | Single model | 0.26551 | 0.99376 | 0.15544 | 0.95292 |
| | DWT | 0.20662 | 0.99976 | 0.12897 | 0.99787 |
| | FT | 0.0096842 | 0.99686 | 0.0057582 | 0.93457 |
| | DWT-FT | 0.039312 | 0.99708 | 0.01169 | 0.97312 |
| ANFIS | Single model | 0.26696 | 0.99369 | 0.15653 | 0.95291 |
| | DWT | 0.20114 | 0.99977 | 0.12044 | 0.99811 |
| | FT | 0.0093793 | 0.99705 | 0.0028397 | 0.98247 |
| | DWT-FT | 0.03928 | 0.99709 | 0.0090655 | 0.98329 |
| SVR | Single model | 0.26844 | 0.99362 | 0.15785 | 0.95137 |
| | DWT | 0.23098 | 0.9997 | 0.13368 | 0.99767 |
| | FT | 0.020093 | 0.99484 | 0.012544 | 0.96885 |
| | DWT-FT | 0.1033 | 0.99532 | 0.083158 | 0.97231 |

**Table 8 Ensemble method results.**

| | RMSE | R |
|---|---|---|
| Single model | 0.75907 | 0.95513 |
| Ensemble model | 0.966634 | 0.99803 |

levels are merged. The scale of data in the combined model is bigger than the single model. Therefore, to compare the results, the correlation coefficient should be considered.

## LIMITATIONS

Despite the good results that the proposed ensemble approach theoretically provides, in practice, price time-series forecasting is faced with challenges that limit their use and, in some cases, make it impossible. Algorithms use methods such as violet, which not only removes noise in the data but also changes the dimensionality of the data. Prices in the real world are accompanied by noise, so removing noise can change the data's nature. Although the use of fuzzy transformations reduces the accuracy of the models, it preserves the data's nature. Also, time-series data are strongly correlated with delay 1 (see related figures in "Data and Results"). It causes the forecasting to be done with one delay phase. To remove this problem, the smoothed price, such as moving average or exponential moving average may be used instead of the original price.

## SUMMARY AND CONCLUSION

In this research, a new approach is presented to identify optimal time delays. Also, for removing the noise from data, the wavelet decomposition has been used. On the other hand, for modeling uncertainty of gas price time-series that influenced by political, economic, social, etc., factors, the fuzzy transform with different membership functions is used. Afterward, the wavelet decomposition and fuzzy transformation are combined for improving the accuracy of forecasting and modeling uncertainty. Finally, the best-selected models are integrated as an ensemble method for obtaining the best accuracy and performance. In this paper, for the time-series of natural gas prices, optimal time delays (1, 2, and 5) were identified by a new proposed approach. Then, wavelet decomposition and fuzzy transform with different membership functions are used in the preprocessing stage. In this study, four models were presented to forecast the natural gas price. In the first model with the input of the original time series, RMSE and R of RBF were 0.1483 and 0.95731 respectively as the best performance. In the second model or wavelet neural networks, the values of RMSE and R of W4-MLP have been obtained 0.00061674 and 1, respectively, as the best performance. In the third model or fuzzy neural networks, the values of RMSE and R of F6-MLP with Triangular type 1 membership function were 0.00010555 and 0.99949, respectively, as the best performance. In the fourth model or wavelet-fuzzy neural networks, the values of RMSE and R of W1-F5-RBF combination have been obtained 0.00011825 and 0.99875, respectively, as the best performance. Finally, by integrating the forecasting results for the original time series of natural gas price, the values of RMSE and R were 0.75907 and 0.95513, respectively. The combination of the forecasting results for time series decomposed by wavelet and fuzzy transforms has been performed, and the values of RMSE and R have been calculated 0.966634 and 0.99803, respectively. All four models, in terms of RMSE and R, are favorable but comparing the four models, the second model considering the correlation coefficient, performed better than the other three models.

### Funding
The authors received no funding for this work.

### Competing Interests
The authors declare that they have no competing interests.

### Author Contributions
- Faramarz Saghi analyzed the data, performed the computation work, prepared figures and/or tables, and approved the final draft.
- Mustafa Jahangoshai Rezaee conceived and designed the experiments, performed the experiments, analyzed the data, performed the computation work, authored or reviewed drafts of the paper, and approved the final draft.

## Data Availability

Data was downloaded from https://finance.yahoo.com/.

Data and MATLAB code are available in the Supplemental Files.

## Supplemental Information

Supplemental information for this article can be found online at http://dx.doi.org/10.7717/peerj-cs.409#supplemental-information.

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
