# Peer review of "An ensemble approach based on transformation functions for natural gas price forecasting considering optimal time delays"

_PeerJ Computer Science, doi:10.7717/peerj-cs.409_

## Round 0.1 · original submission · Major Revisions

I have received reviews of your manuscript from three scholars who are experts on the cited topic. They find the topic very interesting; however, several concerns must be addressed regarding experimental results, hyperparameters choice, computational complexity, mother wavelet choice, decomposition levels selection, contributions to the body of knowledge (gap being-filled), and comparisons with existing methods. These issues require a major revision. Please refer to the reviewers’ comments listed at the end of this letter, and you will see that they are advising that you revise your manuscript. If you are prepared to undertake the work required, I would be pleased to reconsider my decision. Please submit a list of changes or a rebuttal against each point that is being raised when you submit your revised manuscript.

Thank you for considering PeerJ Computer Science for the publication of your research. We appreciate your submitting your manuscript to this journal.

Reviewer 1 ·

Basic reporting

This paper presents a new ensemble approach based on transformation functions for natural gas price forecasting. Then, this approach was applied to the natural gas price dataset. With respect to the obtained results, the new technique is able to provide acceptable results in terms of accuracies. In this reviewer’s opinion, the idea of using a new ensemble approach based on transformation functions for natural gas price forecasting looks promising, but several concerns should be clarified for possible publication. Some comments are listed below.

Experimental design

The experimental results are not convincing. The author should explain how to choose algorithms to evaluate the proposed method.

Validity of the findings

The proposed method consists of several parts. In this reviewer’s opinion, an ablation study is necessary to verify the effectiveness of each part and to analyze the role of the main hyperparameters.

Additional comments

This paper presents a new ensemble approach based on transformation functions for natural gas price forecasting. Then, this approach was applied to natural gas price dataset. With respect to the obtained results, the new technique is able to provide acceptable results in terms of accuracies. In this reviewer’s opinion, the idea of using a new ensemble approach based on transformation functions for natural gas price forecasting looks promising, but several concerns should be clarified for possible publication. Some comments are listed below.
1.How to choose hyper-parameters such as learning rate? How about using the method for dynamic determination of learning rate (e.g using AdaGrad or Adam)?
2.How about the time complexity and space complexity of the proposed algorithm when compared with the previous methods?
3.It is recommended to add some details about how to prevent overfitting.
4.Generally speaking, deep learning methods are time-consuming. Please add some considerations about the computational load of the proposed method. Is GPU based method or Heterogeneous computing been considered while computing gradient?
5.The experimental results are not convincing. The author should explain how to choose algorithms to evaluate the proposed method.
6.What is the state-of-the-art method in this area? What is the performance of the proposed method when compared with the previous state-of-the-art method?
7. The proposed method consists of several parts. In this reviewer’s opinion, an ablation study is necessary to verify the effectiveness of each part and to analyze the role of the main hyperparameters.

·

Basic reporting

- The article was written in English, good grammar, clear and unambiguous
- the article includes the introduction and background which demonstrate the way of work of the research with the relevant literature references
- the structure of the article is in accordance with the format.
-please make the resolution for these figures clearly: Figure1, Figure 2, Figure 3, Figure 4, Figure 5, Figure 6, Figure 7, Figure 8, Figure 9, Figure 11, and Figure 15.
- Please make the line of the graphic for the targets and outputs for Figure 12, Figure 14, Figure 16, Figure 17, Figure 18, Figure 19, Figure 20 are detailed and clear. My suggestion: make the shape of lines are extremely different such as:
______ outputs
-o-o-o-o Targets
so that, we can know the results of outputs and targets very clearly.
- the result of this study are good with the clear definitions and related to the theorems

Experimental design

this research is original and has the aims and scope of this journal

Validity of the findings

This is an interesting study of forecasting and useful.

Additional comments

this study forecast the natural gas price and I recommended accepting this article with minor revisions according to the three areas' comments.

Reviewer 3 ·

Basic reporting

no comment

Experimental design

no comment

Validity of the findings

Fundamental information was not observed, for example
1. The process of selection for the mother wavelet function.
2. Details about data, origin, sampling, nature.
3. The forecast horizon, short-term, long-term.
4. The time lag selection.
On the other hand I have observed some errors, confused or incomplete information, among them:
The autor no define all variables in some equations, for example in 1 and 2, 12, 13
In equation 10 are not included the inputs of the MLP, however in text are described.

Finally, the manuscript contains many techniques, in the preprocessing stage and in forecasting stage, Wavelet+Fuzzy and MLP+RBF+GMDH+ANFIS+SVR, respectively, which accuracy results are similar.
I found a confussion in: figures 1, 2 and 10, section 3.8, because were created various models, however, line 374 and equation 18 do not define properly the work that was done. I find that for me is difficult to identify the contribution of this manuscript.

Additional comments

Fundamental information was not observed, for example
1. The process of selection for the mother wavelet function.
2. Details about data, origin, sampling, nature.
3. The forecast horizon, short-term, long-term.
4. The time lag selection.
On the other hand I have observed some errors, confused or incomplete information, among them:
The autor no define all variables in some equations, for example in 1 and 2, 12, 13
In equation 10 are not included the inputs of the MLP, however in text are described.

Finally, the manuscript contains many techniques, in the preprocessing stage and in forecasting stage, Wavelet+Fuzzy and MLP+RBF+GMDH+ANFIS+SVR, respectively, which accuracy results are similar.
I found a confussion in: figures 1, 2 and 10, section 3.8, because were created various models, however, line 374 and equation 18 do not define properly the work that was done. I find that for me is difficult to identify the contribution of this manuscript.

---

## Round 0.2 · Minor Revisions

Some minor revisions are required. Please improve English grammar and check the spelling. It is also of paramount importance to define the type of wavelet (e.g., Haar, Daubechies, Symlet, among others) that is being used to perform the wavelet decomposition.

Reviewer 1 ·

Basic reporting

The article is now in an acceptable state

Experimental design

The experimental design is reasonable

Validity of the findings

The conclusion is reasonable and sufficient

Additional comments

I have no more opinions

Reviewer 3 ·

Basic reporting

The article presents structure and clarity.

Experimental design

The experimental design has been detailed properly.

Validity of the findings

The findings may be of interest for forecasting researchers

Additional comments

The atention for arbitration observations have improved the article. I consider that there is a significant effort of authors because have used and compared different techniques to reach more accuracy in short term forecasting.

---

## Round 0.3 · accepted · Accept

I am pleased to inform you that your work has now been accepted for publication in PeerJ Computer Science.

Thank you for submitting your work to this journal.